# GC Content and Thermal Stability of Double-Stranded RNA: Fragments of Microsporidia *Vairimorpha ceranae* and *Nosema bombycis* AT-Rich Genes Are Sensitive to Standard Heat Treatment

**DOI:** 10.3390/ijms262110270

**Published:** 2025-10-22

**Authors:** Ruslan R. Fadeev, Sergey A. Timofeev, Igor V. Senderskiy, Viacheslav V. Dolgikh

**Affiliations:** All-Russian Institute of Plant Protection, Podbelskogo 3, 196608 Saint Petersburg, Russia; fadeeff.rusln@gmail.com (R.R.F.); ts-bio@ya.ru (S.A.T.); senderskiy@mail.ru (I.V.S.)

**Keywords:** RNA interference, double-stranded RNA, GC content, *E. coli* HT115 (DE3), bacteria heating, AT-rich genes, thermostability, RNA denaturation, RNA degradation

## Abstract

Heating at 95 °C or boiling *E. coli* HT115 (DE3) cells is often used to extract heterologous dsRNA or kill bacteria, although these temperatures cause dsRNA denaturation and destruction. In this study, we examined the risk of degradation of dsRNA fragments of AT-rich genes at high temperature. The expression of dsRNA fragments of AT-rich genes encoding DNA replication enzymes from the microsporidia *Vairimorpha ceranae* and *Nosema bombycis* in *E. coli* HT115 (DE3) was accompanied by heating the bacteria at 95 °C for 30 min. In contrast to four control fragments with normal GC content, the AT-rich dsRNAs of microsporidia were destroyed by this treatment. The in vitro synthesis and heating of the studied dsRNAs showed the degradation of both microsporidia and control fragments. The thermal degradation of in vitro-synthesized control dsRNA with a normal GC content of 47.6% was prevented by the addition of 2 × YT media, NaCl, or low concentrations of MgSO_4_. This demonstrates the important role of mono- and divalent cations in stabilizing heated fragments and helps explain the preservation of their integrity and RNAi-initiating activity despite the treatment of bacteria at temperatures that denature dsRNA. Feeding Colorado potato beetle larvae with the same in vitro-synthesized dsRNA containing fragments of three *Leptinotarsa decemlineata* genes showed that their thermal destruction was accompanied by a decrease in pest-suppressing activity. No dsRNA degradation was observed at 80 °C or after *E. coli* sonication, and these treatments, as well as increasing cation content, may help to avoid the degradation of heat-sensitive dsRNA.

## 1. Introduction

In 1998, RNAi, a post-transcriptional gene silencing mediated by dsRNA complementary to a target sequence, was first described in the nematode *Caenorhabditis elegans* [1]. Further studies have demonstrated the efficacy of RNAi in specific silencing the transcriptional activity of target genes in other eucaryotic cells, including insect pests [2,3,4], phytopathogenic fungi and viruses [5,6,7,8], pathogens of beneficial insects [9,10,11]. Cost-effective dsRNA biosynthesis for large-scale experiments has been largely achieved through the development of (1) the L4440 vector with a polylinker flanked by two copies of the T7 promoter [12], (2) the RNase III-deficient *E. coli* HT115 (DE3) strain [13], and (3) efficient methods for the extraction and isolation of expressed dsRNA. These methods often include a step of extracting dsRNA from bacteria by heating them at 95 °C or boiling [14,15,16,17,18,19], sometimes in the presence of 0.1% SDS (Table 1). In crude material bioassay experiments, dsRNA-producing bacteria can also be killed by heating at 95–100 °C [17,20,21].

It is well known that dsRNA denatures at 95–97 °C [22], followed by degradation of ssRNA [23,24]. Thus, there is a risk of dsRNA destruction during such heat treatment, and the most convenient objects to study this issue are dsRNA fragments of AT-rich genes, for which low thermal stability can be expected. Among all eukaryotes analyzed [25], the most AT-rich transcripts and CDSs were found in parasitic species of the genus *Plasmodium* (GC content 21.2–25.6%), microsporidia of the genera *Nosema*/*Vairimorpha*, *Hepatospora*, *Enterocytozoon*, *Anncaliia* (24.8–27.8%), protozoans of the genus *Entamoeba* (27.7–27.8), anaerobic intestinal fungus *Pecoramyces ruminantium* (26.8%), parasitic nematode *Strongyloides ratti* (28%), and free-living protozoans of the genera *Dictyostelium* and *Tetrahymena* (27.4–27.7%).

In this study, we demonstrated three main points. (1) The thermal stability of bacterially expressed dsRNA fragments of AT-rich genes encoding the microsporidia *Nosema bombycis* and *Vairimorpha ceranae* DNA replication enzymes is significantly lower compared to control molecules with normal GC content. (2) Mono- and divalent cations present in the bacterial culture medium play an important role in dsRNA stabilization. (3) Thermal degradation of dsRNA fragments is accompanied by a decrease in the efficiency of RNAi.

## 2. Results

### 2.1. GC Content in CDSs of Microsporidia V. ceranae and N. bombycis Genes Encoding DNA Replication Enzymes

In contrast to the mean GC content in human, animal and plant CDSs (about 51%) [26,27], it averaged about 29% in the ten analyzed microsporidia DNA replication genes (Table 2).

Fragments of these genes selected for dsRNA biosynthesis in *E. coli* were also AT-enriched compared to fragments of genes from the Colorado potato beetle *Leptinotarsa decemlineata* and the green peach aphid *Myzus persicae* taken in our experiments as positive controls (Table 3).

Among the studied microsporidia sequences, the fragment and CDS of *N. bombycis* delta DNA polymerase showed the highest GC content. The fragment and CDS of *V. ceranae* epsilon DNA polymerase were found to be the most enriched with AT. Even greater differences between the microsporidia and control gene fragments were found when assessing the total nucleotide content (%) in regions 15 bp or longer that lack adjacent G/C (Table 3, Appendix A).

### 2.2. Heating E. coli HT115 (DE3) Cells Expressing dsRNA Fragments of AT-Enriched Microsporidia Genes at 95 °C Resulted in Their Destruction

As expected, synthesis of *L. decemlineata* and *M. persicae* dsRNAs with 43.4–53.6% GC in *E.coli* followed by heating the bacteria in water at 80 or 95 °C for 30 min, centrifugation, and agarose gel analysis of the extracts did not reveal degradation of the control fragments. Chimeric dsRNA carrying 21–23 bp siRNA sequences predicted by GeneScript and Thermo Fisher Scientific servers in the *V. ceranae* DNA helicase CDS (Appendix A) with 41.3% GC showed similar thermal stability (Figure 1).

In the case of the dsRNA fragments of the microsporidia *N. bombycis* and *V. ceranae* DNA replication genes, their integrity was preserved after bacteria incubation at 80 °C. However, the corresponding dsRNA bands in agarose gel disappeared if bacteria were heated at 95 °C for 30 min (Figure 2 and Figure 3). The dsRNA of the *N. bombycis* delta DNA polymerase gene with the highest GC content (37.6%) and the lowest portion of 15 bp or longer fragments without adjacent G/C (36.4%) was only partially destroyed by this heat treatment (Figure 2).

Extraction of *V. ceranae* topoisomerase II and helicase fragments from *E. coli* at 80 °C in water followed by their incubation at 95 °C for 30 min showed the same dsRNA degradation as in the case of heating of bacterial suspensions (Figure 4a). Since high-temperature lysis of bacterial cells is often performed in the presence of 0.1% SDS [14,15,19], it is important to emphasize that this concentration of the detergent in *E. coli* suspensions did not prevent the degradation of the studied microsporidia dsRNA fragments at 95 °C (Figure 4b).

### 2.3. Effect of Bacteria Heating Time on the Degradation of the dsRNA Fragment of the AT-Rich Gene

To study the rate of dsRNA of AT-rich genes degradation in bacteria upon heating to 95 °C, biosynthesis of the epsilon subunit fragment of *V. ceranae* DNA polymerase in *E. coli* cells was followed by incubation at 95 °C for 2, 5, 10, 15, or 20 min (Figure 4c). Gradual degradation of dsRNA was observed from 5 min of heating, and most of the fragment was destroyed within 10–20 min of incubation.

### 2.4. Bacteria Sonication May Be Used to Efficiently Extract dsRNA Fragments of AT-Rich Genes

Since sonication of *E. coli* HT115 (DE3) cells is one of the most efficient methods for dsRNA extraction [28,29], we investigated the integrity of the *V. ceranae* DNA ligase fragment after such treatment of bacteria. As shown in Figure 4d, sonication of 0.6 mL of bacterial suspension for 10 s three times using a Q700MPX sonicator (QSonica, Newtown, CT, USA) with a Cup Horn at 50% amplitude resulted in efficient dsRNA extraction without its degradation. Thus, this approach can be successfully used to isolate dsRNA fragments of AT-enriched genes.

### 2.5. In Vitro Synthesized dsRNA Fragments Are More Sensitive to High-Temperature Treatment than E. coli-Derived Ones

To compare the thermal stability of *E.coli*-derived and in vitro-synthesized dsRNA, three *L. decemlineata* and *M. persicae* fragments with normal GC content, used as controls in this study, were generated using the MEGAscript™ RNAi Kit (Thermo Fisher Scientific, Waltham, MA, USA) and the HiScribe^®^ T7 High Yield RNA Synthesis Kit (New England Biolabs, Ipswich, MA, USA). Subsequent heating of the synthesized products at 95 °C for 30 min and their analysis on agarose gel showed degradation of the 401 and 869 bp *L. decemlineata* fragments with 43.4 and 47.6% GC (Figure 5), although the same *E. coli*-derived dsRNAs were resistant to this treatment. In contrast, the 679 bp *M. persicae* nuclease gene fragment with a higher GC content (53.6%) demonstrated stability at 95 °C similar to that of bacteria-derived dsRNA.

To assess the stability of the in vitro-synthesized dsRNA of AT-enriched microsporidia genes, *V. ceranae* DNA ligase and topoisomerase II fragments with GC content of 29.3 and 32.5% were generated using the HiScribe^®^ T7 High Yield RNA Synthesis Kit (New England Biolabs, Ipswich, MA, USA). As expected, heating at 95 °C for 30 min destroyed the synthesized dsRNA (Figure 6).

### 2.6. Mono- and Divalent Cations Present in the Bacterial Culture Medium Play an Important Role in Stabilizing Heated dsRNA

To determine the cause of the different thermal stability of in vitro-synthesized and bacteria-derived dsRNA, the control fragment with a 47.6% GC content (*Ld* trimer) was extracted from bacteria at 95 °C and further purified using a standard protocol, including treatment with RNase A, extraction with acidic phenol and chloroform, and ethanol precipitation [29]. Since heating this dsRNA dissolved in water resulted in its degradation (Figure 7a), it was concluded that “stabilizing agents” are present in the bacteria growth culture. For their identification, 19 μL of (1) an extract of untransformed HT115 cells heated at 95 °C, (2) 2 × YT growth medium (1 × and diluted 1:20), (3) NaCl at a concentration corresponding to its content in 2 × YT (86 mM) and diluted 1:10 (8.6 mM), (4) 0.086 mM MgSO_4_ were mixed with 1 μL of a solution containing 1.5 μg of in vitro-synthesized, ethanol-precipitated and water-dissolved *Ld* trimeric dsRNA. Subsequent heating of the mixtures at 95 °C for 30 min showed a stabilizing effect everywhere except for 8.6 mM NaCl (Figure 7b). This result demonstrated that monovalent and, especially, divalent cations present in the growth medium play an important role in stabilizing dsRNA during high-temperature treatment of bacteria.

### 2.7. Thermal Destruction of In Vitro-Synthesized Ld Trimer dsRNA Containing Fragments of Three L. decemlineata Genes Was Accompanied by a Decrease in Pest-Suppressing Activity

To test whether thermal degradation of dsRNA affects the RNAi effectiveness, in vitro-synthesized *Ld* trimer molecules carrying fragments of three *L. decemlineata* genes were used to feed 2nd instar pest larvae. Daily application of dsRNA to potato leaves at a dose of 20 ng per 5 larvae for 7 days revealed a statistically significant reduction in pest mortality if the fragments were preheated at 95 °C for 30 min (Figure 8). In contrast to insect feeding with unheated dsRNA, mortality of larvae that ingested thermally destroyed fragments was not significantly different from that observed for the untreated control.

## 3. Discussion

Rapid degradation of ssRNA at 95 °C or at higher temperatures is a well-known phenomenon. Unlike DNA, ribose in RNA has a hydroxyl group 2′-OH, which can interact with the phosphate group to cleave bonds via transesterification [23]. For instance, mRNAs consisting of 1000 or more nucleotides have an expected half-life of about 1 min or less at 100 °C [24]. In the case of dsRNA fragments, their integrity and RNAi-initiating activity after incubation at 95 °C or even boiling have been confirmed by many authors [14,15,16,17,18,19,20,21]. We have also previously shown that incubation of bacterial suspensions at 95 °C for 20 min allows efficient extraction of dsRNA fragments [29].

In this study, using fragments of ten genes encoding DNA replication enzymes of the microsporidia *N. bombycis* and *V. ceranae*, it was shown that for dsRNA with a relatively low GC content, there is a limit of tolerance to heating at 95 °C. The difference in GC content between the *N. bombycis* delta DNA polymerase fragment, which was partially degraded in our experiments, and the nearest heat-sensitive dsRNA of *V. ceranae* helicase was only 3.8% (37.6 and 33.8%, respectively). More significant difference between these fragments was observed when assessing the proportion of regions (15 bp or longer) without adjacent GC nucleotides (36.4 and 56.2%, respectively). It is possible that a more aggregated distribution of GC nucleotides in the *N. bombycis* delta DNA polymerase fragment contributes to an increase in the thermal stability of dsRNA. Thus, it can be assumed that the stability of dsRNA molecules with a GC content of more than 38% should be sufficient for their efficient extraction from *E. coli* HT115 (DE3) cells at 95 °C. If the GC content in a fragment does not exceed 37%, sonication of bacterial suspensions or other approaches [15] may help to effectively extract dsRNA and avoid their heating at 95 °C or higher temperatures.

In vitro synthesis of the studied dsRNAs followed by heating at 95 °C demonstrated their degradation in commercial solutions without the action of RNases, with the exception of the *M. persicae* nuclease fragment with the highest GC content 53.6%. The different thermal stability of the bacteria-derived and in vitro-synthesized dsRNA fragments of *L. decemlineata* genes was unexpected, since the same plasmids were used for *E. coli* HT115 transformation and amplification of templates for in vitro synthesis. However, further investigation revealed that bacterial extracts and bacterial growth medium contain components that stabilize heated dsRNA. Since centrifuge pellets of HT115 (DE3) cells expressing dsRNA are often loose and viscous, indicating disruption of bacterial integrity and DNA release, their thorough washing can lead to significant loss of dsRNA. Therefore, the resulting bacterial preparations retain a significant amount of bacterial medium. Our experiments demonstrated that its components play a crucial role in stabilizing dsRNA in heated bacteria, and their removal during further purification of the fragments reduces their stability.

It is well-known that the stability of RNA molecules and their folding into stable tertiary structures are extremely sensitive to the concentrations and types of cations [30,31], and Mg^2+^ has been found to preserve the RNA structures better than Na^+^ [32]. In this study, we also demonstrated that Mg^2+^ ions more effectively protect dsRNA from thermal degradation, since significantly higher NaCl concentrations are required to achieve a similar protective effect. Despite the lower efficiency of Na^+^ in stabilizing RNA structure, LB and 2 × YT growth media contain significant amounts of NaCl. Thus, both mono- and divalent cations should play an important role in enhancing the dsRNA thermal stability.

The practical significance of the low thermal stability of dsRNA fragments of AT-rich genes depends on the prevalence of such CDSs in species important for dsRNA treatment. Sequences with GC content below 37% may be widely presented in various pests and bloodsuckers, since the average GC percentage in CDSs of 15 Coleoptera and 42 Diptera species is only 40% [33]. In the case of the honey bee *Apis mellifera*, probably the most important insect species for humankind, the average GC content in genes is 35% [34]. Since we demonstrated that thermal destruction of dsRNA is accompanied by a decrease in RNAi efficiency, GC content should be taken into account for such AT-rich CDSs.

Although the RNAi components Dicer and Argonaute have not been found in the AT-rich genomes of intracellular parasites of the genus *Plasmodium* [35] and fungal opportunists of the genus *Pneumocystis* [36,37], there are at least five studies demonstrating the effect of RNAi on the growth of the microsporidia *V. ceranae*, which causes nosemosis of honey bee *A. mellifera* [38,39,40,41,42]. In the case of *Entamoeba histolytica*, which causes amebiasis in humans, a functional RNAi pathway has also been identified [43], the parasite RISC was characterized [44], and the GC content in the pathogen genome of 24.2% [45] is comparable to that of microsporidia. Cost-effective dsRNA biosynthesis may someday be required to control these pathogens.

Another need to consider the low thermal stability of dsRNA may arise when analyzing their content in samples using qPCR. Denaturation of dsRNA at 95 °C should be performed prior to cDNA synthesis [46], and for dsRNA fragments with low GC content, this may lead to their premature degradation. To date, the possibility of degradation of heat-sensitive dsRNAs during their denaturation for cDNA synthesis remains unexplored.

## 4. Materials and Methods

### 4.1. Analysis of GC Content in the Studied Sequences

DNASTAR’s Lasergene sequence analysis software (version 5.05) [47] was used to calculate the GC nucleotide content in the studied dsRNA fragments and microsporidia CDSs. To assess the uniformity of GC distribution in the studied fragments, the percentage of regions longer than 14 bp that did not contain adjacent G or C was also calculated.

### 4.2. Plasmid Construction

To express dsRNA fragments with normal GC content in bacteria, we used the pRSETa plasmid (Thermo Fisher Scientific, Waltham, MA, USA), previously modified into pRSETRNAi 1 [29,48], with cloned fragments of genes encoding (1) endonucleases of the green peach aphid *M. persicae* (MN257576.1), (2) the b’-COP subunit of the coatomeric complex of the Colorado potato beetle *L. decemlineata* (XM_023172101.1), (3) subunit A of the vacuolar ATPase (XM_023156517.1), subunit 7 (Mov34) of the 26S proteasome (XM_023171125.1), and actin (KJ577616.1) of *L. decemlineata* fused into a heterotrimer [29] (Appendix A). The gene fragments of the *V. ceranae* delta (XP_024330101.1) and epsilon (EEQ83015.1) subunits of DNA polymerase, helicase (XP_024331020.1) and topoisomerase II (XP_024331425.1) were previously cloned into the vector pRSETRNAi 2 [48] (Appendix A).

The fragments of the *V. ceranae* DNA ligase (EEQ82740.1) (Appendix A), *N. bombycis* delta (EOB12369.1) and epsilon (XP_013171816.1) subunits of DNA polymerase, helicase (EOB15277.1), topoisomerase II (EOB12949.1), DNA ligase (EOB13379.1) (Appendix A) were amplified using parasite genomic DNA, Phusion Flash High-Fidelity PCR Master Mix (Thermo Fisher Scientific, Waltham, MA, USA) and the primer pairs listed in Table 4. A chimeric DNA fragment carrying out 21–23 bp siRNA sequences predicted by GenScript’s siRNA Target Finder (https://www.genscript.com/tools/sirna-target-finder, accessed on 6 March 2024) and Invitrogen Block-iT RNAi Designer (https://rnaidesigner.thermofisher.com/rnaiexpress, accessed on 22 February 2024) in *V. ceranae* DNA helicase CDS (Appendix A), was artificially synthesized by the Evrogen company (Moscow, Russia). Gel-purified fragments were cloned into the pRSETRNAi 2 plasmid using *Bam*HI/*Hind*III restriction sites, and the accuracy of cloning was confirmed by sequencing with the T7 rev primer (Table 4).

### 4.3. Synthesis of dsRNA in Bacteria E. coli HT115 (DE3)

The constructed plasmids were transformed into *E. coli* HT115 (DE3) cells at 1700 V using the Electroporator 2510 (Eppendorf, Hamburg, Germany), and single colonies transformants selected on solid LB medium containing ampicillin (0.15 mg/mL) were transferred into liquid 2 × YT medium supplemented with ampicillin (0.1 mg/mL) and tetracycline (0.012 mg/mL). Bacterial cultures were grown to OD_600_ 0.6 and 0.5 mM IPTG (isopropyl-ß-D-thiogalactopyranoside, final concentration) was added. After induction of expression, bacteria were incubated 5 h more, collected by centrifugation at 2500× *g* for 10 min, and stored at −80 °C.

### 4.4. In Vitro Synthesis of dsRNA

MEGAscript™ T7 Transcription Kit (Thermo Fisher Scientific, Waltham, MA, USA) and HiScribe^®^ T7 High Yield RNA Synthesis Kit (New England Biolabs, Ipswich, MA, USA) were used to produce dsRNA in vitro. Templates flanked by two opposite T7 promoters were amplified by PCR using T7 for primer (Table 4), 2 × DreamTaq PCR Master Mixes (Thermo Fisher Scientific, Waltham, MA, USA), and constructed plasmids as templates [45]. The concentration of ethanol-precipitated, washed, and water-dissolved PCR products was measured using an NP80 nanophotometer (Implen, Munich, Germany). 0.5 μg of amplified template was added to 10 μL of the reaction mixture and it was incubated overnight at 37 °C. Synthesized dsRNAs were precipitated and washed in the same way as template DNA, dissolved in RNase-free water to 10 volumes of the reaction mixture before determining the concentration. In vitro synthesized dsRNAs and DNA templates were stored at −80 °C until further use.

### 4.5. Treatment of E. coli Cells and dsRNA

The cell pellets after dsRNA synthesis in *E. coli* were resuspended in water to 1/10 of the bacterial culture volume and 0.2 mL of the suspensions were incubated in 1.5 mL microcentrifuge tubes at 80 or 95 °C for 30 min (unless otherwise specified) using Thermit (DNA-Technology, Moscow, Russia) and TDB-120 (Biosan, Riga, Latvia) solid-state thermostats. The temperature in the tubes was additionally monitored using a mercury thermometer. After centrifugation of the heated suspensions at 14,000× *g* for 10 min, the supernatants were aspirated, and extracted dsRNA was analyzed in agarose gel. In some experiments, supernatants after centrifugation of suspensions heated at 80 °C were additionally heated at 95 °C for another 30 min. Expressed in *E. coli* and extracted at 95 °C *Ld* trimer dsRNA was additionally purified using a standard procedure including RNase A treatment, extraction with acidic phenol and chloroform, precipitation and washing with ethanol [29].

In the case of in vitro synthesis, about 1–1.5 μg of dsRNA in 20 μL of RNase-free water or in the solutions specified in Section 2.6. was heated in 0.2 mL microcentrifuge tubes at 95 °C for 30 min before agarose gel analysis.

Sonication of 0.6 mL of bacterial suspension was performed in 1.5 mL microcentrifuge tubes using a Q700MPX sonicator (QSonica, Newtown, CT, USA) with a Cup Horn at 50% amplitude. Each of the three rounds included 10 s of sonication with the same intervals of tube cooling at 4 °C. After sonication, insoluble debris was pelleted by centrifugation at 14,000× *g* for 10 min and dsRNA in the supernatant was analyzed in an agarose gel.

All experiments with bacteria and dsRNA treatments followed by agarose gel analysis were repeated at least three times.

### 4.6. Analysis of dsRNA in Agarose Gel

The processed dsRNA samples were separated in 1% or 1.5% agarose gel with a dsDNA molecular weight marker and visualized using the fluorescent dye ethidium bromide (3,8-diamino-5-ethyl-6-phenylphenanthridinium bromide).

### 4.7. Feeding of L. decemlineata Larvae with dsRNA, Statistical Analysis and Data Visualization

The 2nd-instar Colorado potato beetle larvae were maintained in an MLR-352 climate chamber (Panasonic, Osaka, Japan). Each replicate was a ventilated Petri dish containing five larvae. Three dishes with 15 larvae served as untreated controls. The same number of insects were fed unheated dsRNA, while four dishes with 20 larvae were fed dsRNA heated to 95 °C. Three round potato leaf disks, each approximately 2 cm^2^ in area, were placed daily in each dish for 7 days. The disks were stacked, and 20 ng of in vitro-synthesized *Ld* trimer dsRNA in 100 µL of water was applied between them. In the control, water was applied to the leaves. An additional piece of untreated potato leaf was added to the dishes overnight.

The survminer R package (version 0.5.1) was used to assess the survival of insects from the untreated control group and those treated with dsRNA and dsRNA heated at 95 °C. The Log-rank test was used to evaluate the difference between the obtained survival curves, and the ggsurvplot function was used to visualize the survival probability. Fisher’s exact test was used to assess the difference in the proportion of surviving and dying insects on day 7 of treatment for pairwise comparisons between RNA-treated insects and the control group.

## Figures and Tables

**Figure 1 ijms-26-10270-f001:**
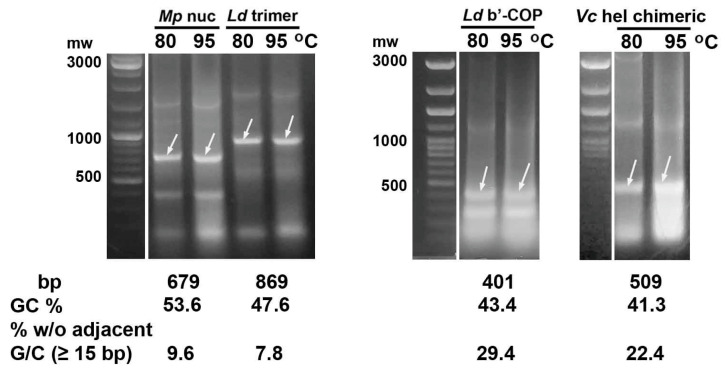
Heating *E. coli* HT115 (DE3) cells after expression of control dsRNA fragments (marked with arrows) of genes encoding *M. persicae* nuclease (*Mp* nuc), *L. decemlineata* trimer v-ATPaseA-mov34-actin (*Ld* trimer), *L. decemlineata* b’-subunit of COPI (*Ld* b’-COP) and fused siRNAs predicted in CDS of *V. ceranae* helicase (*Vc* hel chimeric) with a normal GC content showed their thermal stability at 80 or 95 °C.

**Figure 2 ijms-26-10270-f002:**
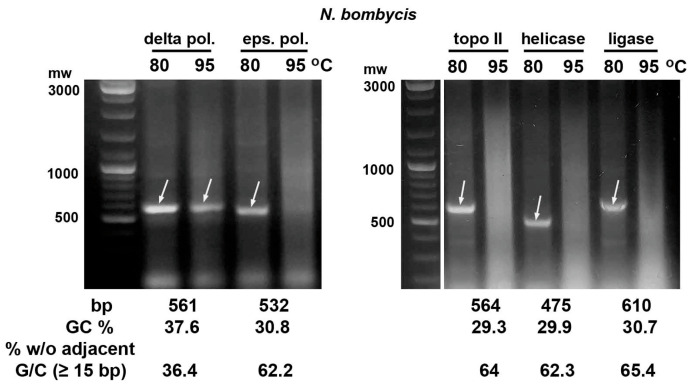
Heating of *E. coli* HT115 cells after expression of dsRNA fragments (marked with arrows) of genes encoding the microsporidium *N. bombycis* epsilon subunit of DNA polymerase (eps. pol.), topoisomerase II (topo II), helicase and ligase with GC content of 29–31% showed their degradation after incubation at 95 °C for 30 min. The fragment of gene encoding the *N. bombycis* delta subunit of DNA polymerase (delta pol.) with 37.6% GC was partially destroyed by this treatment.

**Figure 3 ijms-26-10270-f003:**
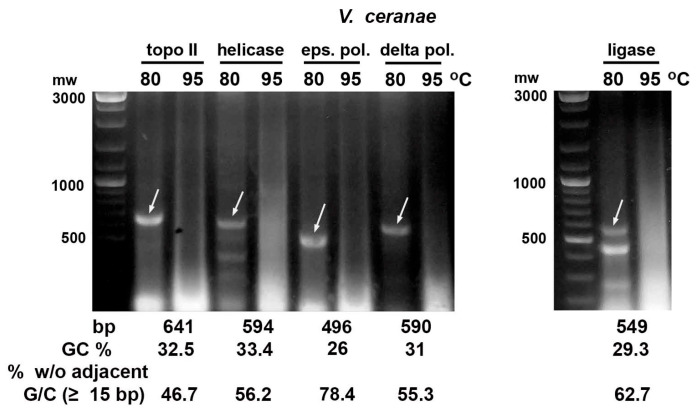
Heating of *E. coli* HT115 (DE3) cells after expression of dsRNA fragments (marked with arrows) of *V. ceranae* DNA replication enzymes with a GC content of 26–33% showed their degradation at 95 °C. Abbreviations as in Figure 2.

**Figure 4 ijms-26-10270-f004:**
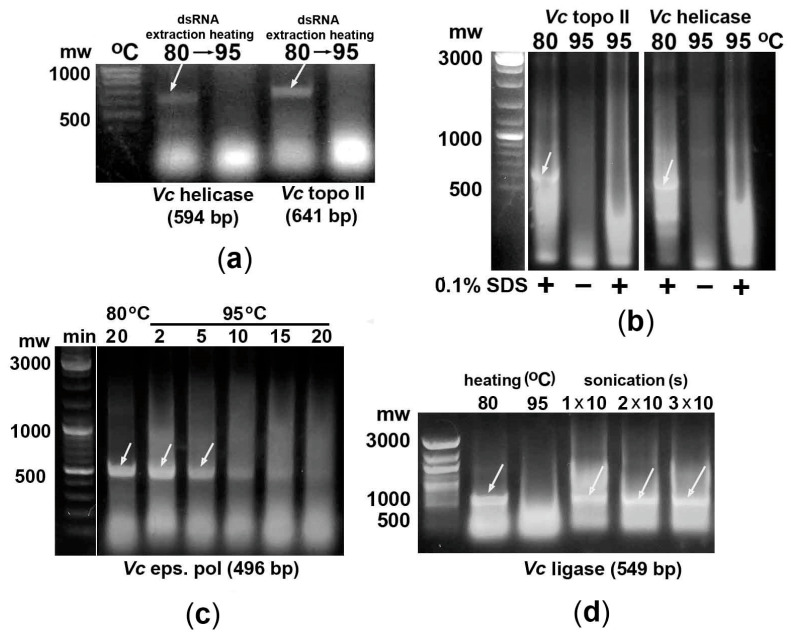
Some peculiarities of high-temperature degradation of dsRNA fragments of AT-enriched genes (marked with arrows). (**a**). Microsporidia dsRNA fragments preliminarily extracted from *E. coli* at 80 °C showed similar degradation at 95 °C, as in the case of heating the bacteria. (**b**). The presence of 0.1% SDS in *E. coli* bacterial suspensions did not prevent the degradation of the studied microsporidia dsRNA fragments at 95 °C. (**c**). Biosynthesis of the epsilon subunit fragment of *V. ceranae* DNA polymerase in *E. coli* cells with different times of their treatment at 95 °C showed gradual degradation of dsRNA. Most of the fragment was destroyed after 10–20 min of heating. (**d**). Similarly to heating of bacterial cells at 80 °C, their sonication allows the extraction of dsRNA fragments of AT-rich genes without their degradation.

**Figure 5 ijms-26-10270-f005:**
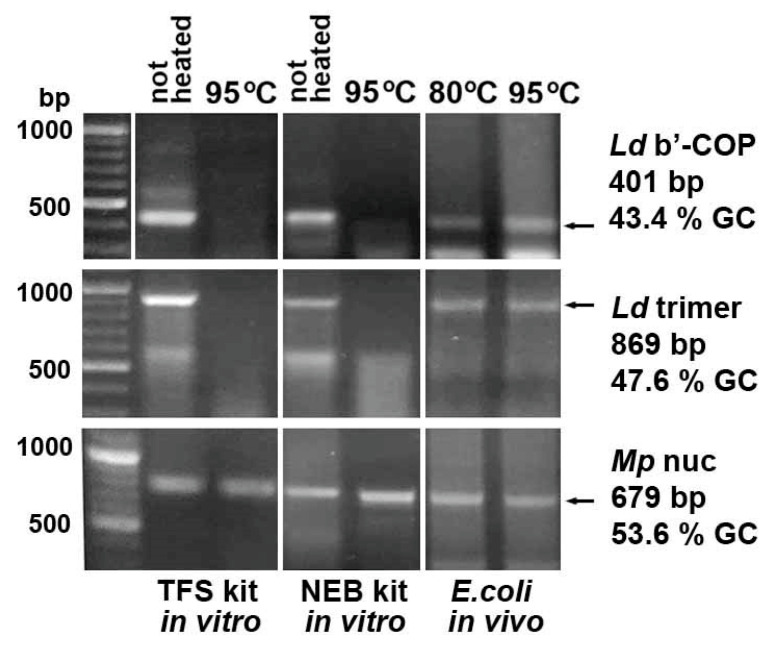
In vitro synthesis of three control fragments followed by heating at 95 °C for 30 min showed degradation of the *L. decemlineata* dsRNA with 43.4 and 47.6% GC contents, whereas the same *E. coli*-derived dsRNAs were resistant to this treatment. In contrast, dsRNA fragment of the *M. persicae* nuclease gene with 53.6% GC content demonstrated thermal stability similar to that of bacteria-derived dsRNA. MEGAscript™ RNAi Kit produced by Thermo Fisher Scientific (TFS kit, Waltham, MA, USA) or HiScribe^®^ T7 High Yield RNA Synthesis Kit produced by New England Biolabs (NEB kit, Ipswich, MA, USA) were used for in vitro dsRNA production.

**Figure 6 ijms-26-10270-f006:**
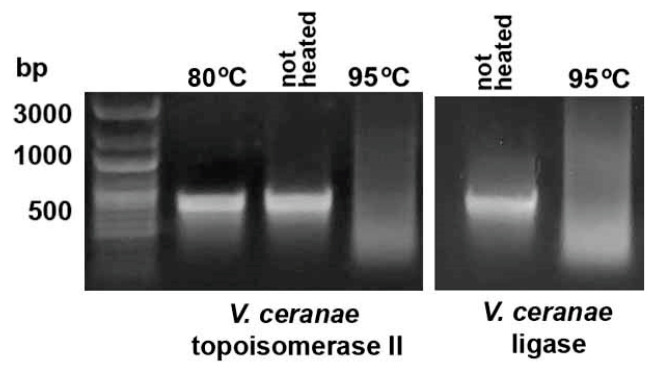
In vitro synthesis of DNA ligase and topoisomerase II fragments of *V. ceranae* with a GC content of 29.3 and 32.5%, followed by heating at 95 °C for 30 min showed the destruction of microsporidia dsRNA.

**Figure 7 ijms-26-10270-f007:**
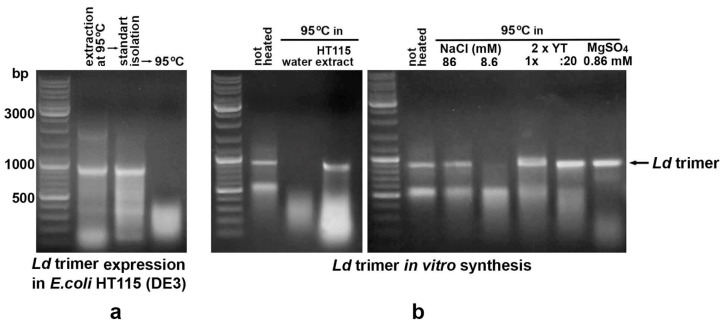
Degradation of *E. coli*-derived, purified and heated at 95 °C *Ld* trimer dsRNA with a GC content of 47.6% (**a**), and thermal stability of its in vitro synthesized form in the presence of 2 × YT growth medium, high NaCl or low MgSO_4_ concentrations (**b**) indicate the important role of mono- and, especially, divalent cations in stabilizing heated dsRNA.

**Figure 8 ijms-26-10270-f008:**
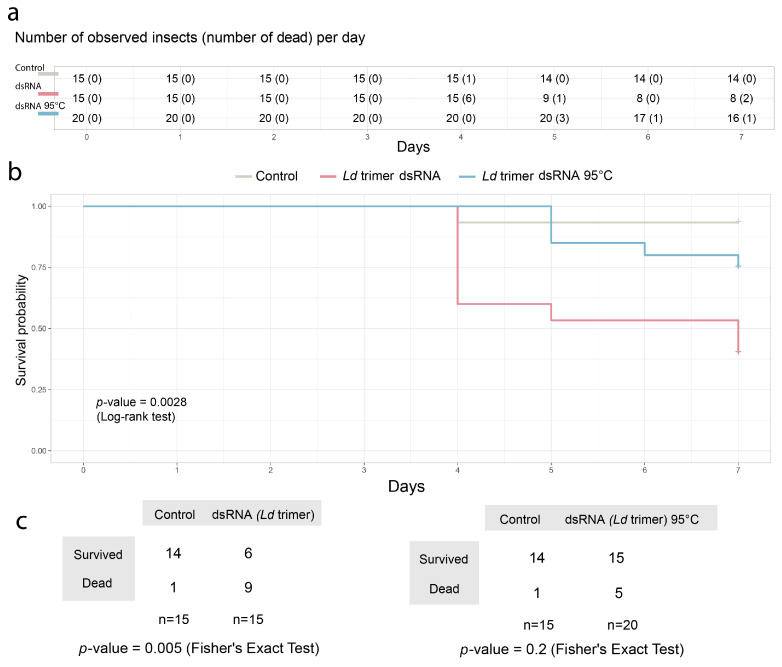
Feeding 2nd instar Colorado potato beetle larvae with dsRNA containing fragments of three *L. decemlineata* genes (*Ld* trimer) showed that its thermal destruction at 95 °C is accompanied by a decrease in pest-suppressing activity. (**a**) Insect survival rates (1) in the untreated control group, (2) in the group treated with *Ld* trimer dsRNA, and (3) in the group fed *Ld* trimer dsRNA heated at 95 °C. The number of insects examined on each study day, as well as the number of dead larvae on the observation day (indicated in parentheses), are presented in the table. (**b**) The survival graph displays the predicted probabilities of insect survival from the three groups over 7 days. The Log-rank test was used to assess differences between survival curves. (**c**) Pairwise comparison of the difference in the proportion of survived and dead larvae at the end of the 7-day observation period, assessed using Fisher s exact test.

**Table 1 ijms-26-10270-t001:** Examples and conditions for high-temperature treatment of bacteria expressing dsRNA.

Bacteria	Temperature (°C)	Heating Time (min)	0.1% SDS	Ref.
*E. coli* HT115 (DE3)	95	2	+	[19]
– *	100	2	+	[14,15]
–	95	10	–	[17,18]
–	100	10	–	[16]
–	100	60	–	[20]
lactic acid bacteria	95	10	–	[21]

*—the same *E. coli* strain.

**Table 2 ijms-26-10270-t002:** GC content in sequences encoding DNA replication enzymes of microsporidia *V. ceranae* and *N. bombycis* and CDSs of animal and plant genomes.

Coding Sequences	GC Content in CDS (%)	Ref.
Human and animal genomes (6 species)	50.7	[26]
Plant genomes (20 species)	50.8	[27]
*V. ceranae*		
delta subunit of DNA pol. (XP_024330101.1)	31.1	this study
epsilon subunit of DNA pol. (EEQ83015.1)	22.8	– *
DNA helicase (XP_024331020.1)	29.6	–
DNA topoisomerase II (XP_024331425.1)	28.9	–
DNA ligase (EEQ82740.1)	28.9	–
*N. bombycis*		
delta subunit of DNA pol. (EOB12369.1)	34.6	–
epsilon subunit of DNA pol. (XP_013171816.1)	28.8	–
DNA helicase (XP_013161385.1)	28.9	–
DNA topoisomerase II (XP_013181718.1)	28.5	–
DNA ligase (EOB13379.1)	28.6	–

*—the sequence was analyzed in this study.

**Table 3 ijms-26-10270-t003:** List of gene fragments from microsporidia and pest species selected for dsRNA biosynthesis in *E. coli* HT115 (DE3).

Gene	Size (bp)	GC (%)	% bp w/o Adjacent G/C *
*N. bombycis*			
delta subunit of DNA pol.	561	37.6	36.4
epsilon subunit of DNA pol.	610	30.7	65.4
DNA helicase	475	29.9	62.3
DNA topoisomerase II	564	29.3	64
DNA ligase	532	30.8	62.2
*V. ceranae*			
delta subunit of DNA pol.	590	31	55.3
epsilon subunit of DNA pol.	496	26	78.2
DNA helicase	594	33.8	56.2
DNA topoisomerase II	641	32.5	46.7
DNA ligase	549	29.3	62.7
siRNAs of DNA helicase (chimeric)	509	41.3	22.4
*M. persicae*			
nuclease	679	53.6	9.6
*L. decemlineata*			
v-ATPaseA-mov34-actin	869	47.6	7.8
b’-COP	401	43.4	29.4

*—the total nucleotide content (%) in regions 15 bp or longer that lack adjacent G/C.

**Table 4 ijms-26-10270-t004:** List of primers used for PCR amplification of *V. ceranae* and *N. bombycis* gene fragments.

Primers	Sequence	PCR-Product Size (bp)
*Vc* ligase *Bam*HI for	atcggaTCCGGAATAAAATCTAGAATTTAC ^1,2^	526
*Vc* ligase *Hind*III rev	cgtaagCTTCTCCGTCTATTACAAAATC	
*Nb* delta *Bam*HI for	TTTGTTATGGATCCTAAGAGAG	539
*Nb* delta *Hind*III rev	gataagCTTTTTTCATGTCTGTCTCATG	
*Nb* epsilon *Bam*HI for	tgcggATCCTGAAAATGCACAAGATAAAG	588
*Nb* epsilon *Hind*III rev	CAATAAAATAGCTTAATAAGCTT	
*Nb* helicase *Hind*III for	atcaagCTTAAAGATTTTAGGAGAAATC	453
*Nb* helicase *Bam*HI rev	TTAAGGATCCTAAATCTTTATAATC	
*Nb* topoII *Bam*HI for	tcaggaTCCTATTGAGATGCACAAGGAAG	542
*Nb* topoII *Hind*III rev	cagaagCTTGTTGAAAATGTTCATCACTAAC	
*Nb* ligase *Hind*III for	actaagCTTTTAGTTAAATTTTTACAAGAG	510
*Nb* ligase *Bam*HI rev	atcggatCCATCAAAATAAAGGCAATCAA	
T7 for	taatacgactcactataggg	variable
T7 rev	ctagttattgctcagcggtgg	

^1^ Restriction *Bam*HI and *Hind*III sites for fragment cloning are noted in gray. ^2^ Sequences complementary to microsporidia gene sequences are indicated by capital letters.

## Data Availability

The original contributions presented in this study are included in the article/Appendix A. Further inquiries can be directed to the corresponding author.

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
