# Peer review of "GC Content and Thermal Stability of Double-Stranded RNA: Fragments of Microsporidia Vairimorpha ceranae and Nosema bombycis AT-Rich Genes Are Sensitive to Standard Heat Treatment"

_ijms, 2025, doi:10.3390/ijms262110270_

Round 1

Reviewer 1 Report

Comments and Suggestions for Authors

The authors identifies a practical but overlooked issue: thermal instability of AT-rich dsRNA when subjected to boiling or 95 °C treatments, which are commonly used in insect molecular biology and RNAi studies. The results are useful for guiding experimental practices (e.g., avoiding boiling, preferring sonication or heating at ≤80 °C). The methodology applied is appropriate but incomplete, the points mentioned below need to be addressed:

  1. The study shows dsRNA degradation biochemically, but does not test the impact on in vivo RNAi efficacy (e.g., reduced gene silencing in insects or microsporidia after heat-treated dsRNA).
  2. The manuscript attributes degradation to AT-rich sequence composition but does not probe the structural basis (e.g., stem–loop formation, melting curves, or secondary structure predictions).
  3. The results lack replicate numbers, statistical significance, or quantification method for “half degraded” claims or other relevant description.
  4. The authors mention about qPCR/cDNA synthesis were linked to thermal stability of dsRNAs, this hypothesis should be supported by direct experimental data.
  5. Did the authors conducted experiments to show whether heat-treated dsRNA indeed loses RNAi efficacy in a model insect system?

Author Response

The authors identified a practical but overlooked issue: thermal instability of AT-rich dsRNA when subjected to boiling or 95 °C treatments, which are commonly used in insect molecular biology and RNAi studies. The results are useful for guiding experimental practices (e.g., avoiding boiling, preferring sonication or heating at ≤80 °C). The methodology applied is appropriate but incomplete, the points mentioned below need to be addressed:

1. The study shows dsRNA degradation biochemically, but does not test the impact on in vivo RNAi efficacy (e.g., reduced gene silencing in insects or microsporidia after heat-treated dsRNA).

In the present R1 version of the manuscript we added data of biotests and their statistical analysis showing that thermal destruction of dsRNA was accompanied by a decrease in its pest-suppressing activity.

2. The manuscript attributes degradation to AT-rich sequence composition but does not probe the structural basis (e.g., stem–loop formation, melting curves, or secondary structure predictions).

Unfortunately, in contrast to microRNA and other ssRNA forms, the secondary structure of dsRNA is not so diverse. However, in the new version of the manuscript, we showed that its stability is highly dependent on the presence of cations in the medium.

3. The results lack replicate numbers, statistical significance, or quantification method for “half degraded” claims or            other relevant description.

We added “All experiments with bacteria and dsRNA treatments followed by agarose gel analysis were repeated at least three times” at the end of section 4.6.

Statistical analysis of bioassay results is described in section 4.8.

Since  the phrase " reduced by at least half " is really not good, we modified section 2.3 making it more accurate:

“To study the rate of dsRNA of AT-rich genes degradation in bacteria upon heating to 95°C, biosynthesis of the epsilon subunit fragment of V. ceranae DNA polymerase in E. coli cells was followed by incubation at 95°C for 2, 5, 10, 15, or 20 minutes (Fig. 4c). Gradual degradation of dsRNA was observed from 5 minutes of heating, and most of the fragment was destroyed within 10–20 minutes of incubation.”

Although developing a method to quantify the thermal degradation of dsRNA is challenging, we attempted to determine the difference in precipitation efficiency between intact and degraded fragments. However, both of them precipitated very well with ethanol. Probably, extraction of dsRNA bands from the gel or densitometry, suggested by both reviewers, may be more effective.

4. The authors mention about qPCR/cDNA synthesis were linked to thermal stability of dsRNAs, this hypothesis should be supported by direct experimental data.

We are currently testing different amounts of two contrasting dsRNAs (GC content of 26 and 53.6%) for cDNA synthesis. However, the results obtained are not yet reliable.

Thus, we replaced

“Although in practice dsRNA use for RNAi there is no need to heat synthesized in vitro molecules, it arises when analyzing their content in samples by qPCR. Denaturation of dsRNA at 95 °C should be performed before cDNA synthesis [41], and for dsRNA fragments with a low GC content this can lead to their premature degradation. In our experiments, we successfully synthesized cDNA and amplified the corresponding PCR product for a dsRNA fragment containing 37.9 % GC nucleotides. However, for dsRNA with a GC content of 28.2%, amplification was not successful (unpublished data).”

with

“Another need to consider the low thermal stability of dsRNA may arise when analyzing their content in samples using qPCR. Denaturation of dsRNA at 95°C should be performed prior to cDNA synthesis [46], and for dsRNA fragments with low GC content, this may lead to their premature degradation. To date, the possibility of degradation of heat-sensitive dsRNAs during their denaturation for cDNA synthesis remains unexplored.”

in the end of Discussion.

5. Did the authors conducted experiments to show whether heat-treated dsRNA indeed loses RNAi efficacy in a model insect system?

 In the R1 version of the manuscript we added data of biotests and their statistical analysis showing that thermal destruction of dsRNA was accompanied by a decrease in its pest-suppressing activity.

Reviewer 2 Report

Comments and Suggestions for Authors

All detailed comments and suggestions have been provided in the attached PDF file for the authors’ consideration.

Comments on the Quality of English Language

The English could be improved to more clearly express the research.

Author Response

1. The current abstract lacks a clearly defined objective. It is not immediately clear why this work was carried out and what specific gap it aims to address. While several experimental results are reported, there is little interpretation or analysis of their significance. In its present form, the abstract reads more like a list of observations rather than a concise scientific message. I would recommend clarifying the main research question, explicitly stating the rationale behind the study, and providing a clear concluding statement that highlights the importance or implications of the findings. Additionally, the writing could be improved by shortening some overly long sentences and ensuring that the final message is easier to follow.

In the  new version of the abstract, we  consider the possibility of destruction of dsRNA fragments of AT-rich genes by heating bacteria at denaturing temperatures and received a positive answer. Other results indicate that (1) mono- and divalent cations of the bacterial culture medium play an important role in stabilizing heated dsRNA, (2) thermal degradation of dsRNA fragments is accompanied by a decrease in the efficiency of RNA interference. Unfortunately,  the limited space of the abstract does not allow to discuss the significance of the data obtained.

In the  new version of the abstract, we  consider the possibility of destruction of dsRNA fragments of AT-rich genes by heating bacteria at denaturing temperatures and received a positive answer. Other results indicate that (1) mono- and divalent cations of the bacterial culture medium play an important role in stabilizing heated dsRNA, (2) thermal degradation of dsRNA fragments is accompanied by a decrease in the efficiency of RNA interference. Unfortunately,  the limited space of the abstract does not allow to discuss the significance of the data obtained.

2. The introduction in its current form is overcrowded with details and reads more like a mixture of background plus results rather than a clear scientific introduction.

We have shortened the introduction in the new version of the manuscript and moved some of the references to the Discussion section.

 A well-structured introduction should guide the reader step by step:

A first paragraph introducing RNAi and the importance of dsRNA.

We have provided a brief description of the discovery of RNAi, its mechanism, and the effectiveness of the approach against various pathogens, pests, and also highlighted key studies important for the development of bacterial dsRNA biosynthesis.

“In 1998, RNAi, a post-transcriptional gene silencing mediated by dsRNA complementary to a target sequence, was first described in the nematode Caenorhabditis elegans [1]. Further studies have demonstrated the efficacy of RNAi in specific silencing the transcriptional activity of target genes in other eucaryotic cells, including insect pests [2–4], phytopathogenic fungi and viruses [5–8], pathogens of beneficial insects [ 9-11]. Cost-effective dsRNA biosynthesis for large-scale experiments has been largely achieved through the development of (1) the L4440 vector with a polylinker flanked by two copies of the T7 promoter [12], (2) the RNase III-deficient E. coli HT115 (DE3) strain [13], and (3) efficient methods for the extraction and isolation of expressed dsRNA.

A second paragraph highlighting the potential problem (thermal stability of dsRNA under standard extraction/processing conditions).

Citing 8 publications and summarizing them in Table 1, we showed that heating bacteria at 95-100 °C is often used to isolate dsRNA or inactivate (kill) bacteria. 

“These methods often include a step of extracting dsRNA from bacteria by heating them at 95 °C or boiling [14–19], sometimes in the presence of 0.1% SDS (Table 1). In crude material bioassay experiments, dsRNA-producing bacteria can also be killed by heating at 95-100 °C [17, 20, 21].”

The next paragraph was slightly modified to make it more focused on the studied issue:

“It is well known that dsRNA denatures at 95–97 °C [22] with following degradation of ssRNA [23, 24]. Thus, there is a risk of dsRNA destruction during such heat treatment, and the most convenient objects to study this issue are dsRNA fragments of AT-rich genes, for which low thermal stability can be expected. Among all eukaryotes analyzed [25], the most AT-rich transcripts and CDSs were found in parasitic species of the genus Plasmodium (GC content 21.2-25.6%), microsporidia of the genera Nosema/Vairimorpha, Hepatospora, Enterocytozoon, Anncaliia (24.8-27.8%), protozoans of the genus Entamoeba (27.7–27.8), anaerobic intestinal fungus Pecoramyces ruminantium (26.8%), parasitic nematode Strongyloides ratti (28%), and free-living protozoans of the genera Dictyostelium and Tetrahymena (27.4–27.7%).”

Unfortunately, the present introduction does not follow this logical flow, making it difficult for the reader to follow the “storyline” of the paper. I strongly recommend that the authors restructure and rewrite the introduction to provide a concise, focused, and progressive narrative that leads naturally to the research question

Unfortunately, it is difficult to write a more detailed introduction due to the lack of any data on the effect of GC content on dsRNA stability.

3. Sentences are long and difficult to be followed!

 They were modified.

4. The fourth paragraph of the introduction is problematic because it presents detailed experimental results (GC values, treatment conditions, and degradation outcomes) rather than introducing the study. This section reads like part of the Results rather than an Introduction. At this point, the authors should simply state the research aim based on the knowledge gap highlighted earlier, rather than providing detailed findings. A concise objective statement would be more appropriate here

 In the new version of this paragraph we very briefly list the most interesting results, pointing to AT-rich genes encoding microsporidia DNA replication enzymes  as the main object of study.

“In this study, we demonstrated three main points. (1) The thermal stability of bacterially expressed dsRNA fragments of AT-rich genes encoding the microsporidia Nosema bombycis and Vairimorpha ceranae DNA replication enzymes is significantly lower compared to control molecules with normal GC content. (2) Mono- and divalent cations present in the bacterial culture medium play an important role in dsRNA stabilization. (3) Thermal degradation of dsRNA fragments is accompanied by a decrease in the efficiency of RNAi.”

5. The presentation of results is generally clear, but the section is overly descriptive. The authors simply report GC content values without offering any analysis or interpretation of their biological significance.

The biological significance of GC content variability across genomes, genes, and CDSs is a very important and interesting topic that could be addressed in the Discussion section. However, we believe it extends beyond the thermal stability of dsRNA.

6. The parameter “% bp w/o adjacent G/C” in Table 3 is not self-explanatory. It should be clarified in the legend or text what this measure exactly represents

As a legend to Table 3, the following has been added: “* - total nucleotide content (%) in regions 15 bp or longer in which adjacent G/C are absent.”

7. The authors should provide some quantitative assessment (e.g., densitometric analysis of band intensity) or, at minimum, comment on the biological significance of the observed degradation

 Same comments to figures 2  and 3.

The authors should consider including densitometric or other quantitative analysis to support statements such as “reduced by half” or “virtually invisible.”

In the present R1 version of the manuscript we added data of biotests and their statistical analysis showing that thermal destruction of dsRNA was accompanied by a decrease in its pest-suppressing activity.

Developing a method to quantify the thermal degradation of dsRNA is challenging and   requires further research. We attempted to determine the difference in precipitation efficiency between intact and degraded fragments. However, both of them precipitated very well with ethanol. Probably, extraction of dsRNA bands from the gel or densitometry, suggested by both reviewers, may be more effective. At this stage of the study, we interpret the data on incomplete degradation of dsRNA very cautiously. In the case of Fig.4c we modified section 2.3 making it more accurate:

“To study the rate of dsRNA of AT-rich genes degradation in bacteria upon heating at 95°C, biosynthesis of the epsilon subunit fragment of V. ceranae DNA polymerase in E. coli cells was followed by incubation at 95°C for 2, 5, 10, 15, or 20 minutes (Fig. 4c). Gradual degradation of dsRNA was observed from 5 minutes of heating, and most of the fragment was destroyed within 10–20 minutes of incubation.”

8. ?????

 Section 2.3 was modified.

9. The thermal treatment conditions (80 or 95 °C for 30 min) are described precisely, but the ambient temperature and the number of biological replicates are not specified. For reproducibility, the number of replicates should be clearly stated.

We believe that room temperature values ​​can be omitted since dsRNA degradation was observed only at 95 C and all dsRNA preparations were always stored on ice after defrosting.

We added “All experiments with bacteria and dsRNA treatments followed by agarose gel analysis were repeated at least three times” at the end of section 4.6.

Statistical analysis of bioassay results is described in section 4.8.

10. The discussion section is descriptive but lacks a clear scientific conclusion. While the authors explain the observed degradation patterns of dsRNA fragments at high temperature, the broader implications of these findings are not addressed. A good discussion should not only interpret the results but also answer the “so what?” question. Specifically:

1. Lack of clear conclusion

2. Missing future perspective

3. Limited broader vision

4. Suggestion

The entire paper focuses on an important phenomenon (the degradation of AT-rich dsRNA under heat), but the discussion does not mention its practical relevance for RNAi-based pest/pathogen control or for improving dsRNA production methods. As a result, the manuscript comes across to the reader as merely an experimental observation rather than providing a genuine scientific insight

In the new version of the manuscript, we have significantly expanded the Discussion section by moving some of the references from the Introduction, adding a discussion of new results, and expanding the discussion of the practical significance of the results obtained.

11. version of DNASTAR Lasergene???

DNASTAR Lasergene 5.05

12. The names of insect species are abbreviated in their first mention (e.g., M. persicae, L. decemlineata). should be written in full at their first occurrence and abbreviated thereafter. I recommend correcting this throughout the manuscript

It was verified 

13. the accuracy of cloning was confirmed by sequencing

It was corrected

14. Delete ,

It was deleted

 15. In the sonication section, it is only mentioned that “10 s sonication with the same intervals of cooling at 4 °C” was applied, but the number of cycles (e.g., 5, 10) is not indicated

It was corrected

Round 2

Reviewer 2 Report

Comments and Suggestions for Authors

The revisions have been made appropriately, and in my opinion, the manuscript is acceptable for publication.